# Re-Thinking Sustainable Development within Islamic Worldviews: A Systematic Literature Review

Odeh Al-Jayyousi [1], Evren Tok [2,*], Shereeza Mohamed Saniff [3], Wan Norhaniza Wan Hasan [3], Noora Abdulla Janahi [4] and Abdurahman J. Yesuf [2]

1 College of Graduate Studies, Arabian Gulf University, Manama 329, Bahrain; odjayousi@gmail.com
2 College of Islamic Studies, Hamad Bin Khalifah University, Ar-Rayyan P.O. Box 34110, Qatar; ajyesuf@hbku.edu.qa
3 Centre for Islamic Development Management Studies (ISDEV), Universiti Sains Malaysia, Penang 11800, Malaysia; shereeza@usm.my (S.M.S.); wanhaniza@usm.my (W.N.W.H.)
4 Innovation and Technology Management Department, Arabian Gulf University, Manama 329, Bahrain; janahinorah@gmail.com
* Correspondence: etok@hbku.edu.qa; Tel.: +974-3312-0327

**Abstract:** Many models of economic growth and sustainable development like circular economy, doughnut economy, and sharing economy were articulated to address the global issues including poverty, climate change, and inequity. However, these models were not informed by traditional value-based worldviews. This systematic literature review aims to gain insights on the different models, practices, and drivers for Islamic sustainable development to inform a new discourse for sustainability. Besides, it intends to define emerging themes in sustainable development and explore the viability for adopting Islamic development models to promote inclusive, pro-poor, and human-centred development. The methodology adopted is systematic literature review to identify sustainability models, practices, and drivers in Islam. Policy recommendations and strategic directions are outlined based on the review.

**Keywords:** sustainable development; models of sustainable development; Islamic sustainable development practice; economic development

## 1. Introduction

Development is a complex process that is intended to induce social, economic, and technological change that is underpinned by good governance, education, and innovation [1]. Development is a vision for many nations, but its realization was impeded by many causes including institutions, governance, and limited ability for innovation. Many schools of economic development viewed progress in terms of gross domestic product (GDP) or human development index [2]. Linear growth theories [3] argued that physical assets and capital are necessary but not sufficient conditions for development.

Islamic development models are underpinned by faith-based practices but is still under experimentation due to the challenge of balancing tradition with modernity [4]. However, the global market crises as manifested in the gap between the rich and the poor, financial crises, COVID-19 pandemic, and climate change, propel thinkers to contribute to a sustainable human civilization by articulating an authentic locally rooted paradigm to overcome current challenges due to market and policy failures. This paper is intended to review literature related to Islam and sustainable development during the period 1990–2021 to identify emerging themes, development models, and knowledge gaps to frame a strategy for new directions for sustainable development. A systematic review of the literature was selected since there is a fragmentation of development literature related to Islamic models. Studies notably address sustainability as a product or a process or an outcome or solution,

but no conceptual or systemic approach was developed to frame sustainability in a holistic manner.

Sidani (2019) [5] explored the relationship between Islam and economic underdevelopment by shedding light on the role of historical legacy, cultural values, and Islamic law on development. Research that explored linkages between faith and development found that religious values may influence and shape people's work ethics, which may not be in line with market rationality [6] and Islamic laws may impede economic progress [7–9] due to emphasis of the role of community and traditions [7]. In contrast, empirical evidence in Malaysia, India, and Ghana showed that Islamic values support and foster development as argued by Noland (2005) [10] and Al-Jayyousi (2016) [11]. Many scholars attributed the state of underdevelopment of Muslim nations (*Ummah*) to the absence of local authentic development models [12–14]. The following section presents an overview of the key Islamic principles and values that inform sustainable development.

Islam, originated in the Arabian Peninsula between the period 610–632 CE, is a monotheistic religion that offers a holistic worldview to cosmos, society, economy, and ecology. The Islamic worldview is based on the notion that God of Islam is the same God as that of Christianity and Judaism. God is the Creator of the universe and humans are created with a mandate to be guardians and stewards (*Khalifa*). Islam is viewed as the final revelation and constitutes a continuum of past major religions. Muslims believe in all prophets including Jesus and Moses and view prophet Muhammad as the last messenger who completed God's final message to humanity. The concept of Oneness of God (*Tawhid*) is a core concept in the Islamic worldview that implies that the universe is governed by an overarching system. The cosmic order, balance, and harmony (*mizan*) is the natural state (*fitra*) that humans are mandated to sustain since they are God's vice-regents (*Khalifa*). *Qur'an* along with Prophet teachings represent the reference code and undisputed scripture to draw rules and judgments. Muslims believe that Prophet Muhammad embodied the ideals of Islamic life. His words and actions represent a complementary source of legal ruling and moral guidance to the *Qur'an* [11,15–18].

Islam stresses the value of consciousness and 'afterlife accountability' (day of judgment), which influence human intertemporal choice and behavior. The *Qur'an* describes all living species as communities of life like humans, hence, humankind is obliged to treat all creation with reverence (*taqwā*), compassion (*rahmah*), and utmost good (*ihsān*). The *Qur'an* guides humankind to moderation, balance, and preservation [11,15,18]. At the economic domain, Islam promotes the protection of public goods and limits individual ownership but prohibits usury (*Riba*) and Islamic banking is based on the notion of zero interest and risk sharing. Sustainable development from an Islamic view is to enable people to lead healthy and responsible lives with moderation. Money has come to be recognized as mere tokens, but the paradox is that money supply expands through debt and the current market model is founded on over-extended debt that fosters consumption patterns that are a direct cause of global ecological collapse [11,15,17]. Alms (*Zakat*) is an obligatory wealth tax that Muslims are required to give to the poor. *Zakat* and trust funds (*Waqf*) provide vital mechanisms for fostering social equity. Hence, sustainable development from an Islamic perspective seeks to establish a balance between the environment, economic and social dimensions [11,15,18].

A fundamental principle of Islamic law is that "matters are evaluated in light of their objectives (*maqāsid*)". At least five essential objectives must be considered, the first is religion (*dīn*), moral values, and ethics; life (*nafs*) is the second prerequisite. Third, a society's posterity (*nasl*) must be safeguarded within secure family relationships. Fourth, reason (*aql*) must be safeguarded to ensure rational and ethical behavior. Finally, rights to property (*māl*) are necessary to enable individuals to secure human dignity and livelihoods [11,15].

Based on the aim of the review, the main research question was articulated as follows: What are the main development models, drivers, and practices of sustainable development that are informed by the Islamic worldview? The significance of this research is to shed light on Islamic sustainability models, practices, and drivers to address market failures

as manifested in poverty, consumerism, climate change risks, pollution, and waste [11]. Aydin (2017) [19] commented on how consumer culture brings less happiness and argued for the imperative for a value-based culture where de-growth, prudence, and frugality (*Zuhd*) is promoted. Nusrate Aziz and Mohamad (2016) [20] highlighted the role of Islamic social business to alleviate poverty. The following section of the paper will outline the methodology. Descriptive analysis will be detailed in section three. Section four will include the content analysis and emerging themes. The final section will include conclusions and future research agenda.

## 2. Methodology

This systematic literature review is based on the framework developed by Gough et al. (2012) [21] and Jesson et al. (2011) [22]. The review is intended to identify relevant models, practices, and drivers of sustainable development from an Islamic perspective. The motivation for adopting a systematic literature review was to set a holistic strategy for sustainability research in the developing world due to the divergence of models and approaches that frame Islamic development models. The process for conducting the systematic literature review include three phases, i.e., review protocol, evaluation, and synthesis as outlined in Figure 1.

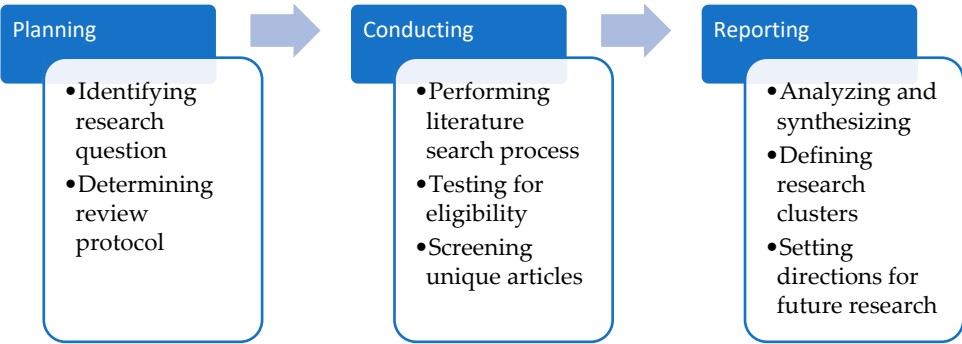

**Figure 1.** The process for the systematic literature review.

The research question guides the search strategy and the review protocol as illustrated in Table 1. The search was based on papers published in English in the Web of Science during 1990–2021. Moreover, other databases were used in the review including Science Direct, Pro Quest Central, EBSCO, and Google Scholar to ensure wider coverage and triangulation. The review used the key words "sustainable development AND Islamic". The rationale for selecting this timeframe is guided by the era that the SDGs process was initiated and marked the global environmental agenda. The global agenda and sustainability discourse was initiated in this era and prior to the SDGs in 1995, which includes the Earth summit and global fora. The review excludes conference proceeding papers, master's theses, doctoral dissertations, textbooks, and unpublished working papers. Web of Science (WoS) (accessible at www.WoS.com (accessed on 3 March 2022) provides the electronic database for the review. WoS is one of the world's largest online databases of peer-reviewed scientific publications and is adequate for SLR [23] and it is regarded as a comprehensive and quality database [24]. Furthermore, it also has a wide network of peer-reviewed literature [25]. The protocol for review is presented in Appendix A.

The evaluation phase identifies and filters through the sources using keyword searches (Topic Search or TS), which provides a combination of searches on journal title, abstract, author keywords, and keywords. The search process was based on the key terms to scan the titles of publications and uses the regions to screen the publication titles, abstract, and keywords for relevant sources using the operator 'AND' to identify and refine logical relationships between terms. The search explored many key words, including Islam, models, practices, and SDGs but the outcome was very limited. After many trials and testing, a final selection of these key words was concluded. The specific search string used is

(TS = (sustainable development)) AND TS = (Islamic)]. Keyword searches through journal titles, abstracts, keywords, and topics yielded 282 articles. Figure 2 shows the compiled body of literature from 1990 to 2021.

**Table 1.** Frequency of articles published in journals.

| Journal's Name | Frequency |
| --- | --- |
| *Sustainability* | 9 |
| *Al-Shajarah* | 7 |
| *Journal of Islamic Marketing* | 7 |
| *Journal of Islamic Accounting And Business Research* | 6 |
| *Advanced Science Letters* | 4 |
| *Global Journal Al-Thaqafah* | 3 |
| *Renewable & Sustainable Energy Reviews* | 3 |
| *Qualitative Research in Financial Markets* | 2 |
| *International Journal of Economics Management and Accounting* | 2 |
| *Arab Law Quarterly* | 2 |
| *Bilimname* | 2 |
| *Etikonomi* | 2 |
| *Religions* | 2 |
| *Jurnal Ilmiah Peuradeun* | 2 |
| *International Journal of Social Economics* | 2 |
| *Turkish Journal of Islamic Economics-Tujise* | 2 |
| *Tarih Kultur Ve Sanat Arastirmalari Dergisi-Journal of History Culture And Art Research* | 2 |
| *International Journal of Intangible Heritage* | 2 |
| *International Journal of Islamic And Middle Eastern Finance And Management* | 2 |
| Other 89 journals | 1 |

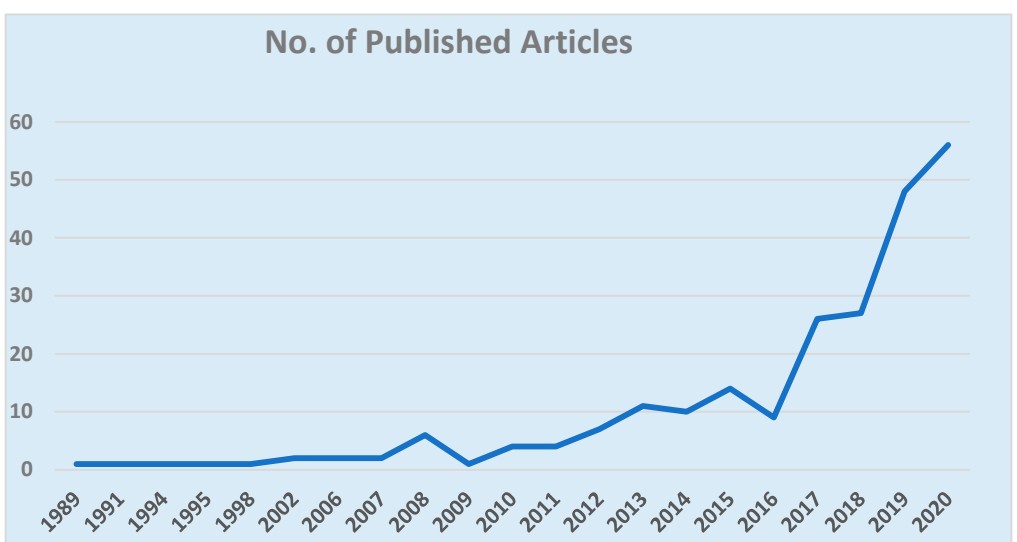

**Figure 2.** Yearly distribution of peer-reviewed published papers used in this review.

In line with the review scope, the research applies cross-referencing and screening to eliminate duplicates and exclude irrelevant sources. The review process involves reading and analysing the key concepts, themes, and findings in the literature in line with the research question. Themes and clusters emerge from this process based on assessing research similarities and differences in terms of regions, core issues, methodologies, and key findings.

The *synthesis phase* entailed combining key findings after the evaluation phase. Guided by the research question, this phase analyses the sustainability models and practices and uses insights from the literature to frame policy directions and recommendation for a transition to sustainability. The articles obtained from this phase were then analysed and

irrelevant articles were excluded. This resulted in 175 eligible articles used for this review as depicted in Figure 3.

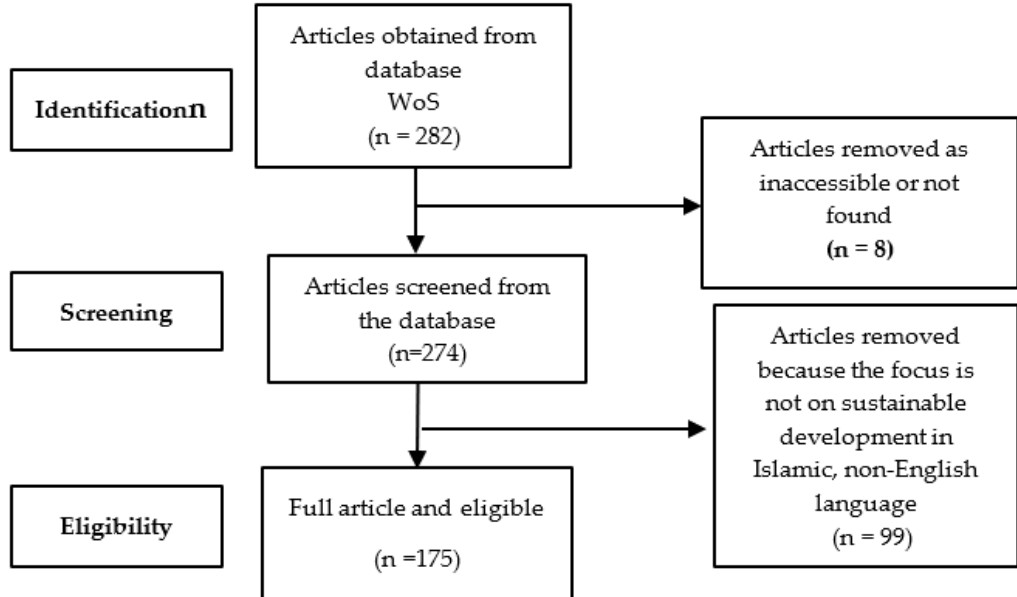

**Figure 3.** Flow diagram for systematic literature review.

Articles in the review were published in a diverse set of journals, including *Sustainability*, *Al-Shajarah*, *Journal of Islamic Marketing*, *Journal of Islamic Accounting and Business Research*, *Advanced Science Letters*, *Renewable & Sustainable Energy Reviews*, and *Qualitative Research in Financial Markets*. Table 1 presents the frequency of articles in published journals.

It was evident that most papers were published in *Sustainability* as shown in Table 1. Many papers used case studies, impact assessment, and indicators as methods for sustainability assessment, as illustrated in Table 2. Besides, many papers focus on ethics and morality, intergenerational equity, cultural diversity, and resource efficiency, as shown in Table 3. Economic, social, and cultural dimensions of sustainability were identified in the review, as shown in Table 4. The main sectors covered in the review were mainly Islamic finance and urban planning, as shown in Table 5.

**Table 2.** Sustainability assessment methods in the review.

| Method | Number of Instances in the Literature |
|---|---|
| Indicators | 4 |
| Regression and neural networks | 1 |
| Sustainability reporting | 4 |
| Green policy analysis | 3 |
| Carbon footprint | 4 |
| Impact assessment | 7 |
| Principle-based framework | 3 |
| Qualitative case study | 11 |
| Zero waste accounting | 1 |
| Multi-critria analysis | 3 |
| Data Envelopment Analysis and efficiency performance | 4 |
| Forecasting and foresight | 2 |

**Table 3.** Islamic sustainability principles in the review.

| Principle | Number of Instances in the Literature |
|---|---|
| Spiritulaity, morality, and ethics | 15 |
| Corporate governance | 1 |
| Social justice | 3 |
| Resource efficiency | 6 |
| Intergenerational equity (Waqf model) | 11 |
| Cultural diversity and co-existence | 7 |
| Ecosystem services and ecological limits | 3 |
| Human security | 1 |

**Table 4.** Sustainability dimensions identified in the review.

| Dimenstion | Number of Instances in the Literature |
|---|---|
| Environmental | 15 |
| Social | 58 |
| Economic | 69 |
| Urban | 13 |
| Institutional and legal | 12 |
| Cultural | 22 |
| Technological | 6 |

**Table 5.** Sustainability Sectors in the review.

| Sector | Number of Instances in the Literature |
|---|---|
| Public administration | 1 |
| Technology | 3 |
| Marketing | 4 |
| Facility management and manufacturing | 2 |
| Tourism | 6 |
| Environment and ecosystem services | 2 |
| Art and culture | 3 |
| Supply chains | 2 |
| Energy and climate change | 5 |
| Water and irrigation | 5 |
| Islamic finance and banking | 28 |
| Microfinance | 6 |
| Social finance | 5 |
| Corporate social responsibility (CSR) | 4 |
| Built environmntal and urban planning | 11 |
| Education | 4 |
| Transport | 2 |

The tables present summarily the diverse forms and approaches on the current state of literature pertaining to Islamic sustainable development in the developing world. The following section outlines descriptive analysis, models, and practices.

## 3. Descriptive Analysis: Themes and Concepts

This study is based on a holistic model-based review [26]. Articles were reviewed to gain insight on models, drivers, and practice of sustainability. Extracted themes, concepts, and dimensions are synthesized as illustrated in Tables 6 and 7. A thematic analysis was performed to identify patterns and emerging concepts [27]. A synthesis of the 175 articles were conducted to extract models, practices, and drivers. Tables 6 and 7 represent the distribution of research articles carried out by country (or region).

**Table 6.** Distribution of sustainability models per country or region.

| Year | Authors | Model | Counrty |
|---|---|---|---|
| 2021 | Glavina, Sofya; Aidrus, Irina; Trusova, Anna [28] | Financial Technology (Fintech) | OIC Countries |
| 2019 | Mahadi, Nur Farhah; Zain, Nor Razinah Mohd; Ali, Engku Rabiah Adawiah Engku [29] | Social finance | Malaysia |
| 2020 | Gedikli, Ayfer; Erdogan, Fatma; Tas, Cihan Yavuz [30] | Sukuk | GCC |
| 2008 | Assi, Eman [31] | Waqf | Palestine |
| 2016 | Solihah, Cucu; Nur, Hilman; Mulyadi, Dedi [32] | Waqf | Indonesia |
| 2021 | Fauziah, Najim Nur [33] | Alternative finance | Indonesia |
| 2021 | Thabith, Muhammed Buhary Muhammed; Mohamad, Nor Asiah [34] | Waqf | Sri Lanka |
| 2016 | Mujani, Wan Kamal; Taib, Mohd Syakir Mohd; Rifin, Mohamad Khairul Izwan [35] | Education | Malaysia |

**Table 7.** Distribution of sustainability practice per country.

| Year | Authors | Practice | Country |
|---|---|---|---|
| 2019 | Hummel, Daniel; Hashmi, Ayesha Tahir [36] | Community development | USA |
| 2020 | Hendratmi, Achsania; Ryandono, Muhamad Nafik Hadi; Sukmaningrum, Puji Sucia [37] | Crowdfunding platform | Indonesia |
| 2019 | Zain, Nor Razinah Mohd; Mahadi, Nur Farhah; Noor, Azman Mohd [38] | Crowdfunding technology | Thailand |
| 2021 | Bhuiyan, Md. Anowar Hossain; Darda, Md. Abud; Hossain, Md. Belal [39] | CSR in Islamic banking | Bangladesh |
| 2020 | Muhammad, Helmi [40] | CSR in microfinance | Indonesia |
| 2012 | Tafti, Saeed Fallah; Hosseini, Seyed Farhad; Emami, Shahnaz Akbari [41] | CSR and Islamic values | Iran |
| 2021 | Danial; Dewi, Nur Sari; Kafrawi [42] | Human capital | Indonesia |
| 2018 | Azmat, Fara; Ferdous, Ahmed; Rentschler, Ruth; Winston, Emma [43] | Art-based initiatives | Australia |
| 2018 | Musahadi [44] | Discourse and communication | Indonesia |
| 2017 | Aziz, Nurul Syaheera; Ismail, Alice Sabrina; Mohidin, Hazrina Haja Bava [45] | Sustainable living | Malaysia |
| 2021 | Ashraf, Muhammad Azeem; Tsegay, Samson Maekele; Ning, Jin [46] | Global citizenship | Pakistan |
| 2020 | Asif, Tahseen; Ouyang Guangming; Haider, Muhammad Asif; Colomer, Jordi; Kayani, Sumaira; ul Amin, Noor [47] | Moral education | China, Pakistan |
| 2019 | Edi, K.; Supriyati; Ramadhan, S. B. [48] | Model enterprise education | Indonesia |
| 2015 | Arshad, Roshayani; Noor, Abdul Halim Mohd; Yahya, Azlan [49] | Islamic social finance | Malaysia |
| 2020 | Julia, Taslima; Kassim, Salina [50] | Green banking | Bangladesh |
| 2019 | Adewale, Adebayo Saheed; Zubaedy, AbdurRaheem Abdul Ganiyi [51] | Islamic finance instruments | Nigeria |
| 2020 | Muneeza, Aishath [52] | Value-based Islamic finance | Maldives |
| 2017 | Sirghani, Mohsen; Mehr, Nasser Delghosha; Dizji, Ozra Hassanzade [53] | Legal review | Iran |
| 2019 | Ercanbrack, Jonathan George [54] | Legal reform | UAE |
| 2020 | Syamlan, Yaser Taufik; Mukhlisin, Murniati [55] | Zero waste accounting | Indonesia |
| 2019 | Begum, Halima; Alam, A. S. A. Ferdous; Mia, Md Aslam; Bhuiyan, Faruk; Ghani, Ahmad Bashawir Abdul [56] | Microfinance for poverty reduction | Bangladesh |
| 2008 | Shunnaq, Mohammed; Schwab, William A.; Reid, Margaret F. [57] | Sustainable tourism | Jordan |
| 2019 | Aman, Jaffar; Abbas, Jaffar; Mahmood, Shahid; Nurunnabi, Mohammad; Bano, Shaher [58] | Sustainable tourism | Pakistan |
| 2017 | Mahmoud, Basma Mohamed Aboelyazeed [59] | Islamic art | Egypt |
| 2019 | Rarasati, A. D.; Bahwal, F. F. [60] | Infrastructure development | Indonesia |

**Table 7.** *Cont.*

| Year | Authors | Practice | Country |
|------|---------|----------|---------|
| 2019 | Alana, Hend A.; Al-hagla, Khalid S.; Hasan, Asmaa E. [61] | Community development | Egypt |
| 2017 | Shibub, Mariam M. T. [62] | Sustainable architecture | Libya |
| 2019 | Kherbouche, Somia; Djedid, Abdelkader [63] | Sustainable cultural tourism | Algeria |
| 2017 | Palupi, Majang; Romadhon, Rizqi W.; Arifan, Nur [64] | Sustainable tourism | Indonesia |
| 2017 | El Amrousi, Mohamed; Elhakeem, Mohamed [65] | Green infrastructure | UAE |
| 2015 | Purnomo, Dwi; Pujianto, Totok; Efendi, Nurfida [66] | Social entrepreneurship | Indonesia |
| 2017 | Cina, Giuseppe; Khatami, Fahimeh [67] | Urban agriculture | Iran |
| 2014 | Rudolff, Britta; alZekri, Muhammad [68] | Traditional knowledge | Bahrain |
| 2018 | Ariesanti, Alia; Irianto, Gugus; Sukoharsono, Eko Ganis; Saraswati, Erwin [69] | Sustainability reporting | Indonesia |
| 2016 | Nobanee, Haitham; Ellili, Nejla [70] | Sustainability reporting | UAE |
| 2020 | Zauro, Nurudeen Abubakar; Saad, Ram Al Jaffri; Ahmi, Aidi; Hussin, Mohd Yahya Mohd [71] | Waqf | Nigeria |
| 2017 | Muller, Dominik M. [72] | Zakat | Malaysia |
| 2021 | Qazi, M. Habib [73] | Education for sustainability | Pakistan |

Research on Islamic sustainable development in the reviewed literature mainly emanate from Asia; Indonesia (20 papers), Malaysia (16 papers), Iran (13 papers), United Arab Emirates (UAE) (7 papers), Pakistan (6 papers), Bangladesh (5 papers), Saudi Arabia (4 papers), Palestine, Egypt, Qatar, Kuwait, India, Nigeria, and GCC Countries (2 papers each) from Africa notably from Nigeria (2 papers) while all the other countries have published one paper as depicted in Figure 4. Sustainability drivers by country or region is presented in Appendix B.

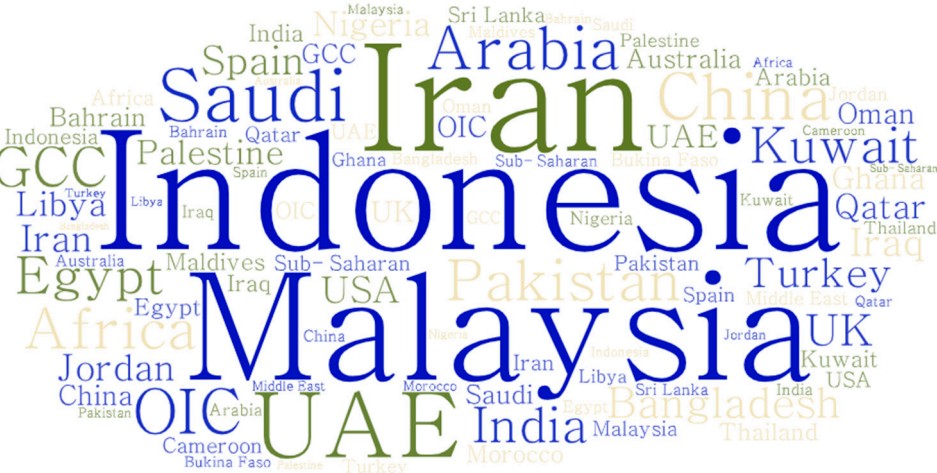

**Figure 4.** A WordArt depicting the countries most researched.

For articles addressing sustainability models, the main models include adoption of fintech, *sukuk* and green *sukuk*, social finance, and *waqf*. Articles that focus on practices encompass domains including community development, crowdfunding, CSR, digital zakat, education for sustainable development (SD), green sukuk, Islamic finance, legal reform, metrics, microfinance, sector, sustainability reporting, waqf, and zakat while articles identified as drivers include case study, criteria, global agenda, and moral principles, as shown in Figure 5.

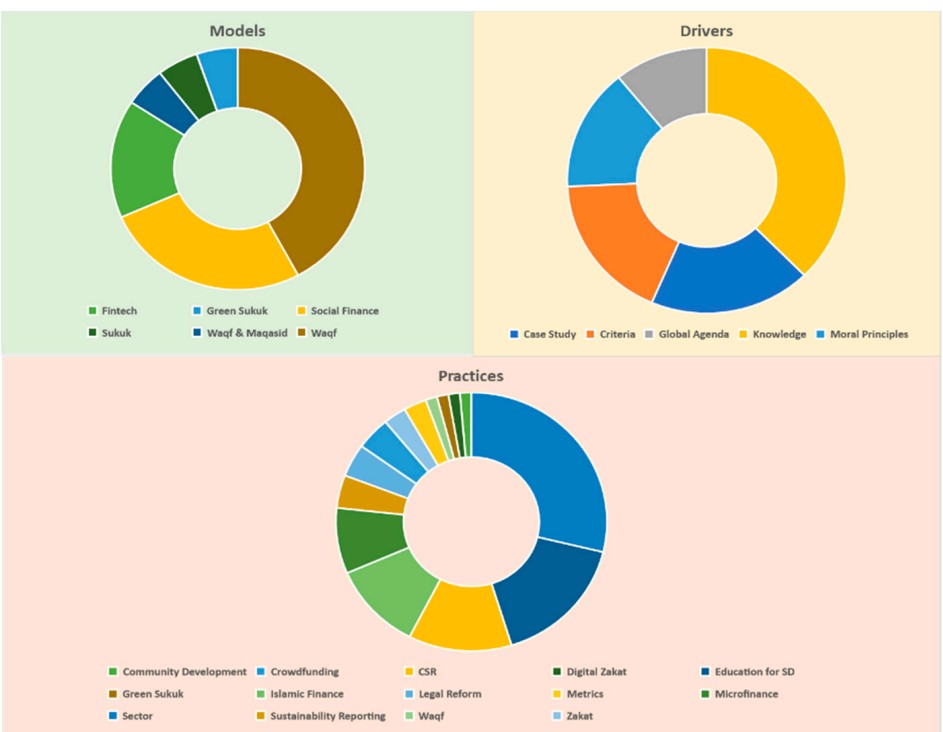

**Figure 5.** Models, drivers, and practices in Islamic sustainability literatures.

This review revealed a set of emerging themes within each of the three key attributes. For the key attribute relating to models, waqf constitutes the main portion as shown in Figure 5 (48%) while social finance constitutes the second component (26%), and fintech (16%). Other models include *sukuk*, and green *sukuk.*

On the other hand, the key sustainability practices include sector-focus research (29%), which covers irrigation efficiency, water management, and ecological restoration. Other practices include projects related to climate and poverty, where the climate–poverty nexus is evident in developing nations since many poor people rely on natural-based development (eco-agriculture, farming, eco-tourism, and forestry) and the water–energy–food nexus. Besides, relevant practices include education for sustainable development (15%), CSR (13%), Islamic finance (11%), and microfinance (8%). The following section will outline the synthesis of the state of research based on the review.

## 4. Current State of Research

Recent studies covered Islamic banking and sustainability [74], which highlighted the need for more systemic analysis to cover all domains of sustainability. Moreover, energy economics was reviewed [75] but it was confined to a specific sector, which inspired this research. Literature review of Islamic sustainable development models, practices, and drivers during 1990–2021 highlights three dimensions of sustainability, i.e., socio-cultural, economic, and environmental. Social sustainability is embedded within universal values and global agenda like SDGs [76], social finance to alleviate poverty [77], CSR and microfinance [40], spirituality and finance [17], cross-cultural values [78], corporate governance [79], education for sustainability [43,80], behavioral factors and Islamic finance [81], sustainable architecture [62], knowledge management [82], *Waqf* system [32], *Zakat* system [72], crowdfunding [38], *Halal* tourism [64], *Halal* supply chains [83], social development, urban planning and eco-cities [84], traditional water management [68], humanitarian aid [85], and co-existence [86].

Framing economic sustainability in Islam highlights the role of Islamic banking in achieving the SDGs [87–89]; sustainability indicators [90]; microfinance and poverty reduction [56]; Islamic finance and SDGs case in Java islands [91]; fintech in Islamic finance [28,92];

green Sukuk [93]; CSR and Islamic finance [94]; barriers for Islamic finance in Oman [95]; Purposes of Islamic law (*Maqasid*) in Islamic finance [96]; Islamic marketing [97,98]; bibliometric review of Islamic microfinance [99]; green *Sukuk* to meet the SDGs and mitigate climate change in Indonesia [48]; *Sukuk* for *Waqf* development in Malaysia [100]; *Zakat* and Islamic banking [101]; harnessing fintech and blockchain in Islamic finance [102]; interest free microfinance to combat poverty [56]; financing higher education in Nigeria using Islamic finance tools [51]; legal reform in Islamic banking [103]; *Halal* tourism [104]; public administration and value-based criteria for performance appraisal [105]; *Waqf* for cultural heritage in Palestine [31]; *Takaful* industry in Bangladesh and Indonesia [106]; *Sukuk* in GCC [30]; risks in Islamic finance in Nigeria [107]; using regression neural network models in Islamic finance to foster sustainability of capital markets [108]; applying digital *Zakat* in economic development in Indonesia [109]; Islamic monetary policy (*Istisnaa*) [110]; Islamic finance model inspired by well-being criterion (*maslaha*) [111]; the SDGs and *Maqasid* [112]; *Halal* food supply chain [113,114]; *Zakah* model for social cause marketing [115]; crowed funding for *Waqf* land model [116]; *Waqf* legal framework in Sri Lanka [34]; traditional water management in Bahrain [68]; energy management in OIC countries [117].

The third cluster in this review shows a set of articles that address environmental sustainability that focus on assessing sustainability in OIC countries using Data Envelopment Analysis (DEA) [118]; energy supply in OIC countries [119]; sustainable transport in OIC countries [120]; institutional development in OIC countries [121]; sustainable tourism strategy in Jordan [57]; environmental protection in OIC countries [122]; sustainability reporting in higher education [69]; sustainability reporting in financial Islamic institutions in Indonesia [123]; carbon footprint for food consumption [124]; zero waste accounting for Islamic institutions to address the SDGs [55]; green banking performance in Bangladesh [50]; sustainable manufacturing [125]; green infrastructure in UAE [65]; *Waqf* forest for the SDGs [126]; mangrove forests in Oman [127]; and green city planning in Qatar [128].

Synthesis of the literature identifies three models of sustainability, i.e., moral-led sustainability, mission-led sustainability, and people-centered sustainability. Consistent with the research question, the remainder of this section presents the findings of models, practices, and drivers in the Islamic sustainable development literature.

a.    **Sustainability Models**

Insights from the literature suggests three main clusters of models for Islamic sustainable development, as summarized by Table 8. The first cluster, moral-led models within the social sustainability, offer a universal perspective that frame global commons where sustainability is contextualized as a platform for cross-cultural learning, and civilizational dialogue that is underpinned by Islamic ethics and values to foster human wellbeing and co-existence [86], trust fund (*waqf*) to promote sustainable development based on cross-cultural values [78], human well-being and public goods (*maslaha*) [111].

In this cluster, sustainable development is contextualized as a unifying narrative for the Oneness of the Creator (*Tawhid*) that represents a platform that fosters harmony, co-existence, and justice, which constitutes the pillars for moral economy. Global commons, intergenerational equity, public interest, and environmental ethics are the main principles within socio-cultural sustainability. The sustainability models informed by the Islamic worldview are underpinned by a constructivist and inter-disciplinary approach, which inform the domain of religion and development as a new discipline embedded in spirituality and finance [17], cross-cultural values [78], and re-definition of good life [11]. Under the socio-cultural sustainability, the review shows human wellbeing models that are underpinned by the objectives of Islamic laws (*maqasid*) through embedding sustainability in culture to serve public goods through fostering education for sustainability [43], sustainable architecture [62], knowledge management [82]; traditional education [80], humanitarian aid [85], and respect for all communities of life [129]. Besides, Islamic finance models were reviewed, which were founded on local values and norms like *Waqf* system [32], *Zakat* system [72], crowdfunding [38], ethical (*Halal*) tourism [64], and *Halal* supply chains [83].

Economic sustainability models, based on the review, were found to contribute to the SDGs in the banking sector in Indonesia [87,88], achieving social justice [89], green *sukuk* for the SDGs [56,93], and Islamic finance and the SDGs case in the Java islands [91]. Measuring and evaluating sustainability were applied at organizational, national, and regional levels, including metrics for sustainability like estimating carbon footprint for food consumption [124] and assessing sustainability in OIC countries using Data Envelopment Analysis [118]. Models and practices reflecting technology for sustainable development were identified like assessing the use of fintech and blockchain in Islamic finance [102]; applying digital Zakat in economic development in Indonesia [109]; blockchain and fintech in Islamic finance [102]. Guided by Islamic law and intent of jurisprudence (*Maqasid*), the review shows linkages to SDGs and [112]; Halal food supply chain [113,114]; Halal tourism [104] (Martin et al., 2020). Specifically, authentic and value-based sustainability models include *Sukuk* (bonds), *Takaful* (insurance), and *Zakat* (Alms) as illustrated in the study of the *Takaful* industry in Bangladesh and Indonesia [106] (Lubaba, et al. 2022); *Sukuk* in GCC [30] (Gedikli, et al. 2020); green *Sukuk* to meet the SDGs and mitigate climate change in Indonesia [48]; *Sukuk* for *Waqf* development in Malaysia [100]; and *Zakat* and Islamic banking [101]. Moreover, *Waqf* models and practices were reviewed that include funding for *Waqf* land model [116]; *Waqf* legal framework in Siri Lanka [34]; and *Waqf*-based model for microfinance [130].

The third dimension, environmental sustainability, includes models framed by regional mandate within Organization of Islamic Conference (OIC) countries, including the study of energy supply in OIC countries [119]; sustainable transport [120]; institutional development [121]; energy management [117]. Metrics and tools for sustainability were identified like green growth policy in Saudi Arabia (Jung, 2018); green banking performance in Bangladesh [50] (Julia and Kassim, 2019); sustainability reporting in higher education [69]; sustainability reporting in financial Islamic institutions in Indonesia [123]; sectoral green policies like green infrastructure in UAE [65]; and sustainable manufacturing [125].

Other models include nature-based solutions through investing in ecosystem services to support sustainability like *Waqf* forest for the SDGs [126], and planting mangrove forests in Oman [127]. Besides, urban sustainability was documented in the review as the case of standards for green city planning in Qatar [128]. The main models within environmental sustainability include sustainable urbanization, nature-based solutions, and education for sustainable development. Overall, conceptually and at the policy and strategic domain, moral-led sustainability models offer an authentic, multi-cultural, and value-based approach to sustainability, while, empirically, the mission-led models (within economic and environmental sustainability) are characterized as nature-based, community-based, outcome-based, and right-based help in localizing sustainability. The following section outlines sustainability models, practices, and drivers.

**Table 8.** Islamic sustainable development models in reviewed articles.

| Dimension | Instance | Overview | Models of Islamic Sustainable Development |
|---|---|---|---|
| **Socio-cultural** | Framing the SDGs withinuniversal values and global agenda like SDGs [76]. | Outlines global dimensions of sustainability, which includes poverty, climate change, co-existence, and human rights. | Islamic sustainable development models founded on global commons, i.e., intergenerational equity, intrinsic value of nature, eco-dimensionality, and harnessing sustainability as a platform for dialogue between civilizations [78]; good life [11]; and co-existence [86]. |

**Table 8.** *Cont*.

| Dimension | Instance | Overview | Models of Islamic Sustainable Development |
|---|---|---|---|
| | Articulating value-based sustainability tools based on social Islamic finance [77]. | Proposes a new approach and modality of Islamic finance and governance like Waqf, Zakat, and Halal operations. | Exploring local models of sustainability founded on corporate governance [79], behavioral factors for adopting Islamic finance [81], *Waqf* system [32], *Zakat* system [72], crowdfunding [38], *Halal* tourism [64], Halal supply chains [83]. |
| | Mainstreaming sustainability in education, CSR, and key sectors and services [40]. | Analyses the rationale and practice for mainstreaming religion in development to inform authentic local models of sustainable development. | Embedding sustainability in culture to serve public goods through education and infrastructure, i.e., education for sustainability [43], sustainable architecture [62], knowledge management [82], urban planning and eco-cities [84], traditional education [80], traditional water management [68], and humanitarian aid [85]. |
| | Embedding drivers of sustainability based on morality and values [17]. | Highlights the value for ethics to promote people-centred development models guided by objectives of Islamic laws (*maqasid*). | Sustainability as a platform for fostering human wellbeing and co-existence [86], trust fund (*waqf*) to promote sustainable development based on cross-cultural values [78], human well-being (*maslaha*) [111]. |
| **Economic** | Highlighting the role of Islamic banking in achieving the SDGs. | Addresses the role of Islamic finance in contributing to sustainability. | The role of banks in Indonesia in achieving sustainable development goals [87], Islamic finance to support the SDGs [88], achieving social justice [89], and green *Sukuk* for SDGs [56,93], Islamic finance and the SDGs case in the Java islands [91]. |
| | Defining sustainability indicators and barriers [90] and behavioral intentions. | Defines indicators like ecological footprints and barriers for sustainability including human factors. | Metrics for sustainability like carbon footprint for food consumption [124]; aassessing sustainability in OIC countries using Data Envelopment Analysis [118]; public administration and value-based criteria for performance appraisal [105]; local knowledge in water in Bahrain [68]; barriers for Islamic finance in Oman [95]. |
| | Adoption of fintech in Islamic finance [28,92]. | Explores the use of financial technology like blockchain and fintech in Islamic finance. | Harnessing fintech and blockchain in Islamic finance [102]; applying digital Zakat in economic development in Indonesia [109]; blockchain and fintech in Islamic finance [102]. |
| | Islamic law and intent (*Maqasid*) and legal reform in Islamic banking [103] and services like food and tourism. | Embeds the *Maqasid* model and *Halal* concept in service sectors to achieve sustainable development. | Reference to purposes of Islamic law (*Maqasid*) in Islamic finance [96]; the SDGs and *Maqasid* [112]; *Halal* food supply chain [113,114]; *Halal* tourism [104]. |
| | Islamic finance model inspired by well-being criterion (*maslaha*) [111] including the *Sukuk* model, *takaful*, and Islamic monetary policy (*Istisnaa*) [110]. | Overviews models of social Islamic finance like *Takaful*, *Zakat*, and *Sukuk*. | Islamic social investment [108]; *Takaful* industry in Bangladesh and Indonesia [106]; *Sukuk* in GCC [30]; CSR and Islamic finance [94]; green *Sukuk* to meet the SDGs and mitigate climate change in Indonesia [48]; *Sukuk* for Waqf development in Malaysia [100]; Zakat and Islamic banking [101] (Rosman, et al., 2019); *Sukuk* in GCC [30]. |

**Table 8.** *Cont.*

| Dimension | Instance | Overview | Models of Islamic Sustainable Development |
|---|---|---|---|
| | Reviving *Waqf* and *Zakat* models for achieving sustainability like *Waqf* for cultural heritage in Palestine [31]. | Illustrates domains and enablers for applying *Waqf* and *Zakat* model for sustainable development. | *Waqf* as a public system (eun, 2019); Islamic social investment [108]; *Zakat* model for social cause marketing [115]; crowfunding for *Waqf* land model [116]; *Waqf* legal framework in Siri Lanka [34]; *Zakat* and Islamic banking [101]; *Waqf*-based model for microfinance [130]; financing higher education in Nigeria using Islamic finance tools [51]; legal reform in Islamic banking [103]. |
| **Ecological** | Regional scale of sustainability including environmental protection in OIC countries [122]. | Regional dimension of sustainability is addressed in energy and transport in OIC countries. | Study energy supply in OIC countries [119]; sustainable transport in OIC countries [120]; institutional development in OIC countries [121]; energy management in OIC countries [117]. |
| | Greening tools like zero waste accounting for Islamic institutions to address the SDGs [55]. | Addresses green policies, metrics, and indicators for sustainability like sustainability reporting and green accounting. | Green banking performance in Bangladesh [50]; sustainability reporting in higher education [69]; sustainability reporting in financial Islamic institutions in Indonesia [123] |
| | Promoting green industries like sustainable manufacturing [125]. | Highlights the role of green infrastructure in green economic growth. | A transition to sustainability is driven by sectoral policies like green infrastructure in UAE [65]. |
| | Fostering nature-based solutions like mangrove forests in Oman [127]. | Evaluates the value of ecosystem services in sustainable development. | Investing in natural capital to support sustainability like Waqf forest for SDGs [126]. |
| | Examining urban sustainability like green city planning in Qatar [128]. | Overviews the pillars for green city planning and conditions for sustainable urbanism. | Green city planning in Qatar [128]. |

Moral-led sustainability models offer a universal perspective of cross-cultural learning and consciousness and highlights the spiritual nature of the universe where "everything is alive, intelligent, and articulate". The moral economy model is articulated within the Islamic sustainable development worldview within core Islamic principles, i.e., oneness of the creator, humans as trustees, and harmony and balance in creation [11]. The human mandate is to serve as guardians and trustees (*Shahed*) to fulfil the divine trusteeship (*Amanah*), protect all forms of life (*Ihsan*), achieve justice and equity (*Mizan*), respect for all communities of life, and secure the balance and natural state (*Fitra*) of the universe. This worldview shapes human knowledge, attitude, and practices of sustainability. Specifically, these models are informed by the Islamic worldview and are based on a constructivist and inter-disciplinary approach to sustainability as reflected in linkages between spirituality and finance [17], cross-cultural values [78,105], and objectives of Islamic laws [129]. Besides, moral-based models include Islamic finance schemes like *Waqf* system [32], *Zakat* system, crowdfunding, and ethical faith-based models like *Halal* tourism [64] and *Halal* supply chains [83]. The review highlights the methodological aspect of exploring the meaning and scope of sustainability within a local and inter-disciplinary approach [17,78,111].

Mission-led sustainability models provide conceptual and policy frameworks to set direction for a transition towards a sustainable future through policy design, economic reform, Islamic finance, and regional sectoral policy in OIC countries. This model calls for a transformative strategy to apply and scale up Islamic development models, including Islamic financing to achieve the SDGs [87,88], green Sukuk for achieving the SDGs [56,93];

Islamic finance and the SDGs in the Java islands [91]. Value-based sustainability models include *Sukuk* in GCC [30]; green *Sukuk* (Hati, et al., 2019); *Sukuk* for *Waqf* development in Malaysia [13]; *Zakat* and Islamic banking [101]; *Waqf* land model [116]; *Waqf*-based model for microfinance [130]. This cluster includes a regional dimension within OIC to assess sustainability in OIC countries using Data Envelopment Analysis [118], harnessing financial technology for sustainable development [102]; and digital *Zakat* [109]. Besides, mission-oriented sustainability models are founded on the purposes of jurisprudence (*Maqasid*), the SDGs and *Maqasid* [112]; *Halal* food supply chain [113,114]; and *Halal* tourism [104].

People-centred sustainability models provide best practices and metrics to augment environmental sustainability. This model considers a sustainable city as a unit of analysis to achieve sustainable urbanism. Human-centred development in this model includes ecological restoration, nature-based solutions, investing in ecosystem services, and revival of arid land and wasteland. The review shows that sustainability is framed in metrics and tools for sustainability like green banking performance in Bangladesh [50]; sustainability reporting in higher education [69]; sustainability reporting in financial Islamic institutions in Indonesia [123]. Other models include nature-based solutions through investing in ecosystem services to support sustainability like *Waqf* forest for SDGs [126], and planting mangrove forests in Oman [127]. Urban sustainability was documented in the review as the case of green growth policy and standards for green city planning in Qatar [128]. Overall, the main models within environmental sustainability include sustainable urbanization, nature-based solutions, and education for sustainable development.

b.   **Sustainability Practices**

Overall, six practices, as summarised in Table 9, account for sustainability in Islam based on the review. These include legal reform, sustainability metrics, technology for sustainable development, education for sustainable development, nature-based solutions, and Islamic finance tools.

**Table 9.** Sustainability practices in reviewed articles.

| Sustainability Practices | Overview | Examples of Application |
|---|---|---|
| Legal reform and governance | Models founded on institutional reform and good governance | • Islamic law and legal reform in Islamic banking [103]<br>• Exploring models based on corporate governance [79] |
| Metrics and indicators for sustainability | Domains for policy reform and new models of governance. | • Carbon footprint for food consumption [124]<br>• Measuring sustainability in OIC countries using Data Envelopment Analysis [118]<br>• Sustainability reporting in financial Islamic institutions in Indonesia [123]<br>• Green banking performance in Bangladesh [50] |
| Technology for sustainable development | Technology-driven models and tools for sustainability | • Fintech and blockchain in Islamic finance [102]<br>• Digital Zakat in economic development in Indonesia [109] |
| Sectoral focus like microfinance and Education for sustainable development | Education for sustainability and sector-based sustainability | • Education for sustainability [43]<br>• Sustainable architecture [62]<br>• Halal food supply chain [113]<br>• Halal tourism [104]<br>• Traditional water management [68]<br>• Urban planning and eco-cities [84]<br>• Traditional education [80]<br>• Sustainable manufacturing [125] |

**Table 9.** *Cont.*

| Sustainability Practices | Overview | Examples of Application |
|---|---|---|
| Nature-based solutions | Investment in ecosystem services and natural capital | • Mangrove forests in Oman [127]<br>• Waqf forest for SDGs [126] |
| Islamic finance models and tools | Microfinance, Islamic bonds, Islamic social finance | • Waqf system [32]<br>• Zakat system [72]<br>• Sukuk for Waqf development in Malaysia [100]<br>• Zakat and Islamic banking [101]<br>• Sukuk in GCC [30] |

c. **Sustainability Drivers**

The review shows five drivers for sustainability, as summarised in Table 10, which range from global agenda, moral principles, knowledge creation, and criteria and standards.

**Table 10.** Sustainability drivers in reviewed articles.

| Sustainability Practices | Overview | Examples of Application |
|---|---|---|
| Global agenda | Drivers for sustainability to support SDGs. | • Islamic finance to support SDGs [88]<br>• Achieving social justice [89]<br>• The practice of green Sukuk for SDGs [93] |
| Moral principles | Sustainability driven by moral principles and ethics. | • Sustainability and cultural diversity [129]<br>• African traditional religions and faith [76]<br>• Moral principles for sustainable development [131]<br>• Sustainability and theology [17]<br>• Western and Islamic values for sustainability [78]<br>• Social harmony and sustainability [132] |
| Knowledge | Sustainability informed by knowledge. | • Barriers to Islamic finance in Oman [95]<br>• Energy status in OIC countries [119]<br>• Renewable energy strategy in OIC countries [133] |
| Criteria | Standards and metrics for sustainability. | • Framework for sustainability performance [90]<br>• Public administration performance [105]<br>• Carbon footprint for food consumption [124]<br>• Social Islamic banking [134] |
| Case study | The demonstration or validation of sustainability. | • Islamic banking and quality of life [91]<br>• Authentic marketing and branding [135]<br>• Attitude formation and e-banking [136]<br>• Behavior intentions and use of Islamic banking [81]<br>• Best practice in corporate governance [137]<br>• Attitudes towards tourism development [138]<br>• Local knowledge and disaster risk reduction [126]<br>• Sustainable marketing and events [97] |

This review contributes to the sustainable development paradigm by framing authentic cultural models and practices of sustainable development informed by Islamic values. The review shows a set of models that are characterized as moral-led, mission-driven, and people-centered models. Inspired by Islamic worldviews, the review sheds light on a set of sustainability practices, i.e., cross-cultural learning, policy reform, technology-driven, and nature-based solutions. The review shows a focus on policy reform and governance, which highlights the imperative to build organizational capabilities and conditions conducive for

a transition to sustainable development. However, limited literature exists on foresight and sustainability, which implies that many countries are dealing with pressing current issues and limited time is devoted to future orientation and strategic shifts. A synthesis of sustainability practices reveal that moral-led sustainability practices provide a prism for cultural learning and social transformations in the socio-technological regime. On the other hand, models farmed within mission-oriented sustainability and people-centered models (nature-based, community-based, outcome-based, and right-based) set the trajectory for the value for local solutions based on actions and co-management of natural resources and technology-driven sustainability.

## 5. Future Directions for Research

Global challenges including poverty, COVID-19, climate change, and conflicts are propelling humanity to make a transition in the global governance and sustainability. This implies the need for a deeper analysis of global system dynamics models. Wellbeing economy and doughnut economy were conceptualized to address the planetary boundaries and new metrics and indicators to measure sustainability beyond GDP should be developed. Many countries in the Muslim world (represented in the 58 countries in the Organization of Islamic Conference (OIC)) embark on transitions plans for sustainability and economic diversification. Balancing tradition and modernity, or technology and culture, poses challenges for reform and progress. The arguments in this article offer new insights for socio-technological innovation inspired from local culture [4,11]. With the OIC countries experiencing strategic shifts in business and economic models, it is imperative that scholarship informs new models and practices for sustainable development that embody unity within diversity

Methodologically, the body of literature favors conceptual frameworks and case studies with some interests in statistical analysis. The review recommends multi-methodologies to capture the multi-disciplinary dimensions of sustainable development and Islam. The review recommends further analysis of the OIC region involving innovation systems, energy transitions and sustainability, policy analysis, phenomenology and ethnography, and grounded theory. Theoretically, grounding sustainability studies in the literature cover a broad range of disciplines. However, there is a need to develop a research agenda on mission-oriented policies to transform OIC countries to set new directions for economic development and green economy. Hence, embedding Islamic values within the sustainability paradigm introduces contextual, authentic, and human-centered development models. Conceptually, the review also aspires to provide new insights on research priorities, knowledge gaps, and critiques to dominant logic of sustainability practice. There are also possibilities to frame a new narrative and discourse for sustainable development inspired by Islamic worldviews. The next section outlines future directions for research.

### a.    Mission-Oriented Sustainability Policies

The review shows some research oriented to technology for development including digital technology in finance [102] and digital *Zakat* [109]. However, the digital economy offers opportunities for deep transformations in sustainable agriculture, mobility, health, and education. Thus, this review recommends studies to address the enablers and conditions for developing mission-oriented eco-innovation policies to address systemic problems across sectors and disciplines like the water–energy–food nexus. This review shows a set of practices and policy initiatives in specific sectors such as sustainable architecture [62], *Halal* food supply chain [113], *Halal* tourism [104], and sustainable manufacturing [125]. However, the disruption of supply chains during the COVID-19 pandemic in 2020 propelled many nations to rethink their national priorities in terms of supply chains, public services, and infrastructure. Research on new business models to secure human livelihood and sustainability is imperative. Moreover, it is recommended to study innovation systems and strategies for regional collaborations and networks in OIC by framing mission-oriented policies to achieve the SDGs and foster collaborative innovation networks. Research on the climate change and poverty nexus is vital since it offers new possibilities for alleviating

poverty and ensuring human security. Solutions in terms of technology, innovation, markets, and policy reforms require a holistic approach for R&D to enable developing countries to catch up and leapfrog technology. Policy innovation labs should be explored to address wicked public policy problems and to enable the design of human-centric solutions.

b.    **Circular and Sharing Economies**

The digital era is defining a new trajectory for smart development in governance, mobility, housing, logistics, business, and learning. However, what is lacking is the notion of human empathy and spirituality that creates a balance between humans and technology. Research is recommended on harnessing emerging technologies to create wealth through crowd financing to globalize *Zakat* and *Waqf* models worldwide to achieve the SDGs, including targets related to climate change, poverty alleviation, and energy security. Digital democracy and re-construction of cities impacted by war is a new opportunity for responsible development. Social media and technological innovations can be harnessed to transform developing countries to co-create accountable and transparent governance models to re-construct cities worldwide in the aftermath of conflicts.

c.    **Nature-Centered Development**

Overall, the literature shows coverage of different sectors including education, tourism, manufacturing, energy, and transport. Nonetheless, notably absent in the review is representation from sustainability science, green technology, energy transitions, ecological restoration, biomimicry, climate change, the water–energy–food nexus, and R&D. It will be valuable to harness value-based and ethical eco-innovations like *Waqf* (trust fund) and green Sukuk models augmented by crowd financing and digital technology.

It is imperative for OIC nations to contextualize local rooted and value-based development models inspired from Islamic worldviews like trust fund (*Waqf*) and protected areas (*Hima*) as authentic and indigenous models for localizing the SDGs. Moreover, since most poor people live in OIC countries, it is vital to rethink consumption and production patterns and redefine progress and happiness in terms of human empathy and de-growth (*Zuhd*) or living lightly on the earth for making a transition to a moral, sharing, and circular economy. This deep transformation propels developing nations to chart a variety of development pathways and to re-invent and develop their own authentic sustainable development models and to embrace a wellbeing and moral economy that is founded on the local values, such as the principle of public interest, diversity, co-existence, and moderation.

## 6. Conclusions

Many countries in the Muslim world (represented in the 58 countries in the Organization of Islamic Conference (OIC)) embark on transitions plans for sustainability and economic diversification. Balancing tradition and modernity poses challenges for reform and progress; the arguments in this article offer new insights for socio-technological innovation inspired from local culture. With the OIC countries experiencing strategic shifts in business and economic models, it is imperative that scholarship informs new models and practices for sustainable development.

The review shows a set of models that are characterized as moral-led, mission-driven, and people-centered models. Moral-led sustainability models embody cross-cultural learning and consciousness of the spiritual nature of the universe and is articulated within Islamic core principles, i.e., Oneness of the creator, humans as trustees, and harmony and balance in creation. The moral-led sustainability represents a prism for refining human perspectives of the purpose, value, and meaning of sustainable development. Another dimension identified in the review is mission-led sustainability models, which provides conceptual and policy frameworks to set the direction for a transition towards a sustainable future through policy innovation, economic reform, Islamic finance, and clean technology. This model calls for a transformative vision and agenda that are compelling and unifying to develop collective action. Besides, mission-led sustainability model entails sectoral collaboration in order to develop socio-technological innovation for sustainable development.

The review finds that mission-led sustainability models are manifested in *Sukuk*, *Zakat*, and *Waqf*, which are founded on community-based natural resources governance, a right-based approach, and technology for sustainable development.

People-centered sustainability models are reflected in action and best practices, which include ecological restoration, nature-based solutions, and investing in ecosystem services. The review shows that sustainability is framed in both present action and outlook, including renewable energy, water conservation, sanitation, and health care. Overall, the moral-led sustainability models offer a local and multi-cultural approach to sustainability, while mission-led and people-centered models (nature-based, community-based, outcome-based, and right-based) set path and direction for sustainability. Overall, five practices account for sustainability approaches based on the review, i.e., legal reform, sustainability metrics, technology for sustainable development, education for sustainable development, nature-based solutions, and Islamic finance tools.

This review contributes to the sustainable development paradigm by framing local models of sustainable development informed by Islamic values and principles. Inspired from Islamic worldviews, the review sheds light on a set of sustainability practices, i.e., cross-cultural learning, policy reform, technology-driven, and nature-based solutions. The review shows a focus on policy reform and governance that highlights the imperative to build organizational capabilities and conditions conducive for a transition to sustainable development. However, limited literature exists on foresight and sustainability, which implies that many countries are dealing with pressing current issues and limited time is devoted to future orientation and strategic shifts. A synthesis of sustainability practices reveal that moral-led sustainability practices provide a prism for cultural learning. On the other hand, models farmed within mission-oriented sustainability and people-centered models (nature-based, community-based, outcome-based, and right-based) set the trajectory for the value for local-solutions based on actions and co-management of natural-resources and technology-driven sustainability.

The review recommends multi-methodologies to capture the multi-disciplinary dimensions of sustainable development and Islam. Moreover, the review recommends further analysis of the OIC region involving innovation systems, energy transitions, and sustainability by applying policy analysis, phenomenology, and grounded theory. However, there is a need to develop a research agenda on mission-oriented policies to transform OIC countries to set new directions for economic development and progress. Overall, embedding Islamic values within the sustainability paradigm introduces authentic and human-centered development models, which in turn offer a moral and value-based approach to sustainability.

**Author Contributions:** Conceptualization, formal analysis and synthesis of findings, O.A.-J.; conclusion and future research, E.T.; methodology and descriptive analysis, S.M.S.; resources and data analysis, W.N.W.H.; data visualization, N.A.J.; data analysis, editing, and formatting, A.J.Y. All authors have read and agreed to the published version of the manuscript.

**Funding:** This research received funds in part from (1) the Ministry of Higher Education Malaysia for Fundamental Research Grant Scheme with Project Code: FRGS/1/2019/SSI10/USM/03/1; and (2) NPRP grant #12C-0804-190009, entitled SDG Education and Global Citizenship in Qatar: Enhancing Qatar's Nested Power in the Global Arena, from the Qatar National Research Fund (a member of the Qatar Foundation).

**Institutional Review Board Statement:** Not applicable.

**Informed Consent Statement:** Not applicable.

**Data Availability Statement:** Not applicable.

**Conflicts of Interest:** The authors declare no conflict of interest.

# Appendix A

**Table A1.** Systematic Literature Review Protocol.

| Review Element | Description | Focus of the Review |
|---|---|---|
| Purpose | Aim of the literature review | The purpose of the review is:<br>• To review models, drivers, and practices of sustainable development that are informed by Islamic values. |
| Search strategy | Course of action or plan to inform the search process for the review | The search strategy for the review involves:<br>• Using keywords to search specified databases based on screening and exclusion criteria. |
| Search strings | Combination of keywords to conduct the search | The search strings for the review are:<br>• "Sustainable development" AND "Islamic" |
| Databases | Independent online database with citation data and indexes of scholarly writings | The database used in the review is:<br>• Web of Science (WoS) |
| Screening and inclusion criteria | Conditions for selecting and including review sources | The screening criteria for the review are as follows:<br>• Empirical and theoretical peer-reviewed journal articles during 1990–2021<br>• Journal articles are inaccessible or not found<br>• Research on "sustainable development" models, drivers, and practices based on Islamic values |
| Exclusion criteria | Conditions for omitting publications during the review process | The exclusion criteria for the review are as follows:<br>• Duplicates<br>• Theses, dissertations, textbooks, and unpublished working papers<br>• Non-English papers<br>• Not relevant or does not meet the criteria |

# Appendix B

**Table A2.** Distribution of Sustainability Drivers per Country or Region.

| Year | Authors | Driver | Country |
|---|---|---|---|
| 2020 | Ghoniyah, Nunung; Hartono, Sri [87] | Global agenda (SDGs) | Indonesia |
| 2013 | Choiruzzad, Shofwan Al Banna; Nugroho, Bhakti Eko [139] | Economic growth | Indonesia |
| 2014 | Magd, Hesham A. E.; McCoy, Mark P. [95](Magd and McCoy, 2014) | Finance development | Oman |
| 2013 | Gabbasa, Mohamed; Sopian, Kamaruzzaman; Yaakob, Zahira; Zonooz, M. Reza Faraji; Fudholi, Ahmad; Asim, Nilofar [119] | Energy supply | OIC |
| 2011 | Sopian, Kamaruzzaman; Ali, Baharuddin; Asim, Nilofar [133] | Renewable energy transition | OIC |
| 2021 | Furlan, Raffaello; Sinclair, Brian R. [128] | Green city | Qatar |
| 2020 | Butt, Marghoob S. [122] | Environmental protection | OIC |
| 2020 | Esmaeli, Zohreh Ali; Kheiri, Bahram; Farahbod, Farzin [135] | Authentic marketing | Iran |
| 2017 | Tok, M. Evren; O'Bright, Ben [140] | Spatial planning | Sub-Saharan Africa |
| 2019 | Mohammadi, Nastaran Keshavarz; Sayyari, Aliakbar; Farshad, Aliasgar; Jahanmehr, Nader; Siddiqi, Sameen; Taghizadeh, Rahim; Dye, Christopher [141] | Public Health | Iran |

**Table A2.** *Cont.*

| Year | Authors | Driver | Country |
|------|---------|--------|---------|
| 2015 | Deng, Renee Rabiam; Aziz, Siti Amalina Abdul; Fadhillah, Asmaul Husna Hans; Osman, Ismah; Rosnan, Herwina; Alwi, Sharifah Faigah Syed [136] | Human behavior | Malaysia |
| 2019 | Pitchay, Anwar Allah; Thaker, Mohamed Asmy Mohd Thas; Azhar, Zubir; Mydin, Al Amin; Thaker, Hassanudin Mohd Thas [81] | Human behavior | Malaysia |
| 2021 | Zainal, Nurazilah; Bakri, Mohammed Hariri; Hook, Law Siong; Zaini, Syahrir; bin Ab Razak, Mohd Faizal [142] | Money markets | Malaysia |
| 2021 | Hamidi, Luthfi; Worthington, Andrew C. [134] | Social Islamic banking | Indonesia |
| 2017 | Ab Wahab, Mastura [143] | Values and work behavior | Malaysia |
| 2019 | Meimand, Sajad Ebrahimi; Mardani, Abbas; Khalifah, Zainab; Nilashi, Mehrbakhsh; Ismail, Hairul Nizam; Skare, Marinko [138] | Human attitudes | Iran; Malaysia |
| 2015 | Sapri, Maimunah; Ab Muin, Zafirah; Sipan, Ibrahim; Adjei-Twum, Anthony [144] | Facility management | Malaysia |
| 2019 | Abbas, Jaffar; Hussain, Iftikhar; Hussain, Safdar; Akram, Sabahat; Shaheen, Imrab; Niu, Ben [82] | Innovation | Pakistan |
| 2022 | Jatmiko, Wahyu; Azizon, A. [145] | Values-led development | OIC |
| 2017 | Mensi, Walid; Hammoudeh, Shawkat; Al-Jarrah, Idries Mohammad Wanas; Sensoy, Ahmet; Kang, Sang Hoon [146] | Risk and energy economics | UAE |
| 2019 | Hejazi, Moeine Ossadat; Sarbakhshian, Behnam [147] | Moral principles | Iran |
| 2020 | Habibi, Kyoumars; Hoseini, Seyedeh Maryam; Dehshti, Majid; Khanian, Mojtaba; Mosavi, Amir [148] | Environmental | Iran |
| 2020 | Durugbo, Christopher M.; Al-Jayyousi, Odeh R.; Almahamid, Soud M. [149] | Innovation | GCC |
| 2020 | Jaelani, Aan; Setyawan, Edy; Aziz, Abdul; Wahyuningsih, Nining; Djuwita, Diana [97] | Islamic marketing | Indonesia |
| 2011 | Abdullah, Mohd Mustafa Al Bakri; Ma'Radzi, Akmal Hadi; Saleh, Noor Akmal Mohamad; Kamal, Syazni Zainul; Yaacob, Noorulnajwa Diyana [150] | Innovation | Malaysia |
| 2019 | Oudah, Reem Fuad; Ragab, Tarek; Shokry, Mohammed [151] | Urban design | Saudi Arabia |
| 2020 | Wang, Shanyong; Wang, Jing; Li, Jun; Zhou, Kaile [152] | Values and behavior | China |
| 2018 | Ibrahim, Yusnidah; Ahmed, Iftekhar; Minai, Mohd Sobri [153] | Microfinance | OIC |
| 2020 | Mim, Nusrat Jahan [85] | Values | Bangladesh |
| 2020 | Damari, Behzad; Heidari, Alireza; Bonab, Maryam Rahbari; Moghadam, Abbas Vosoogh [154] | Criteria | Iran |
| 2020 | Pamuk, Aysil Coskuner; Tastemir, Ibrahim Agah; Arpacioglu, Umit [155] | Green design | Turkey |
| 2019 | Spierings, Niels [86] | Diversity | Middle East; North Africa |
| 2016 | ElDegwy, Ahmed E.; Khali, Essam E. [156] | Energy | Saudi Arabia |
| 2021 | Kabir, K. Habibul; Aurko, Shafqat Yasar; Rahman, Md Saifur [117] | Water reuse | OIC |
| 2020 | Thaher, Rehab A.; Mahmoud, Nidal; Al-Khatib, Issam A.; Hung, Yung-Tse [157] | Risk management | Palestine |
| 2021 | Ali, Tahir; Paton, Douglas; Buergelt, Petra T.; Smith, James A.; Jehan, Noor; Siddique, Abubaker [121] | Disaster management | Pakistan |
| 2021 | Costa, Joana; Pita, Mariana [158] | Entrepreneurship | Selected Islamic Countries * |
| 2015 | Azmi, Ilhaamie Binti Abdul Ghani; Ismail, Sharifah Hayaati Binti Syed; Basir, Siti Arni Binti; Norman, Azah Anir Binti; Yusof, Raja Jamilah Binti Raja [105] | Value-based administration | Malaysia |
| 2019 | Jan, Amin; Marimuthu, Maran; Isa, Muhammad Pisol bin Mohd Mat; Shad, Muhammad Kashif [90] | Sustainability indicators | Malaysia; Saudi Arabia; UAE; Iran Kuwait |
| 2019 | Hidayatullah, A. F.; Della, N., V; Elfrida, N.; Haq, D.; Arikhah [124] | Carbon footprint | Indonesia |
| 2021 | Haleem, Abid; Khan, Mohd Imran; Khan, Shahbaz [83] | Supply chain | India |
| 2017 | Motalebi, Mehdi; Khosravi, Hassan [159] | Criteria | Iran |

**Table A2.** *Cont.*

| Year | Authors | Driver | Country |
|---|---|---|---|
| 2021 | Ghai, Rahul [160] | Legal mandate | India |
| 2018 | El Amrousi, Mohamed; Paleologos, Evan K.; Caratelli, Paolo; Elhakeem, Mohamed [84] | Green city | UAE |
| 2022 | Alhammadi, Salah [161] | Islamic finance | Kuwait |
| 2019 | Rosman, Romzie; Haron, Razali; Othman, Nurul Balqis Mohamed [101] | Value-based | Malaysia |
| 2022 | Lubaba, Saeeda; Ahmad, Abu Umar Faruq; Muneeza, Aishath [106] | Value -based | Bangladesh; Indonesia |
| 2013 | Roldan-Canas, Jose; Fatima Moreno-Perez, Maria [162] | Local knowledge | Spain |

\* Egypt, Malaysia, Indonesia, Turkey, Iran, Morocco, Burkina Faso, Cameroon, Kazakhstan, Lebanon, Jordan, Saudi Arabia, United Arab Emirates, and Qatar (selected according to the OIC countries).

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
