# Peer review of "Re-Thinking Sustainable Development within Islamic Worldviews: A Systematic Literature Review"

_sustainability, doi:10.3390/su14127300_

Round 1

Reviewer 1 Report

Dear authors, 

Thanks for the opportunity to review your manuscript titled „Re-thinking Sustainable Development within Islamic Worldviews: A Systematic Literature Review“.

I have a few remarks that you hopefully find helpful to further improve your manuscript:

  • Timing of sample: You should outline more clearly why you have chosen 1990 as a starting point and why this is reasonable.
  • In the introduction the clear need for conducting a SLR is lacking. There is a lot description of why the topic is relevant - fine - but the link to WHY a SLR must be conducted needs to be more clearly articulated. Otherwise it would be also reasonable to conduct an empirical study. 
  • The research question is first mentioned under methodology. But it should be placed in the introduction and can also be used to clearly connect it to justify SLR as the correct methodology for the study.
  • Figure 1 is confusing. It seems that all three steps are just half of the way of a SLR as the graphic stops with half a circle. Why? Just for graphical reason?
  • Regarding the search terms: why was „Islamic“ and not „Islam*“ used - with this approach you would neglect studies that e.g. use the term „Islam“. Same goes for „sustainable development“ why not „sustainab*“ to include „sustainability“? Please justify or adjust.
  • Why was only WoS used as a database? Ok it is a very comprehensive one but a good review should never just rely on one database. Include others at least for triangulation of results now.
  • The protocol can be put into the appendix.
  • According to your sampling funnel, 31 studies could not be found or were inaccessible - this is ca.. 15% of the identified studies in the first step. Maybe that can be reduced through other databases as  recommended before.
  • What about cross-referencing? At the moment there is no details on this and I fear that some relevant studies are left out due to the too narrow focus on one database only.
  • A major point is the delineation from existing literature reviews in the field (e.g. Islamic banking sustainability: A review of literature and directions for future research (2017) or Energy economics in Islamic countries: A bibliometric review (2021)). There should be a paragraph summarizing existing findings of those review studies and how they differ from your study. This would also underline and strengthen the need for the study which I recommended earlier.
  • Now there are way too many tables in the manuscript which disturbs the reading flow heavily. I would suggest to streamline and put many of them into an appendix or delete.

I hope this comments help to further improve your research. Good luck!

Author Response

Dear Editors

We are deeply grateful to the reviewers for taking the time to review our manuscript. We sincerely appreciate all valuable comments and suggestions which helped us to improve the quality of the article. We have given below our responses and explanations point by point for all the comments and suggestions given by the respected reviewers.

We hope that our manuscript will be acceptable for publication in Sustainability.

Respectfully,

Evren Tok, PhD

RESPONSE TO REVIEWER 1

Comment:

Timing of sample: You should outline more clearly why you have chosen 1990 as a starting point and why this is reasonable.

RESPONSE:

The rationale for selecting the start year 1990 is the fact that the global agenda and sustainability discourse was initiated in this era and prior to SDGs in 1995. This includes Earth summit and global fora. We tried to do a search in 1980 but the results were very limited that do not justify to start in 1980.

Comment:

In the introduction the clear need for conducting a SLR is lacking. There is a lot of description of why the topic is relevant - fine - but the link to WHY a SLR must be conducted needs to be more clearly articulated. Otherwise, it would be also reasonable to conduct an empirical study. 

RESPONSE:

SLR was selected since there is a fragmentation of development literature related to Islamic models. Many studies were addressing sustainability as a product or a process or an outcome or solution. No conceptual or systemic approach was developed to frame sustainability in a holistic manner.

Comment:

The research question is first mentioned under methodology. But it should be placed in the introduction and can also be used to clearly connect it to justify SLR as the correct methodology for the study.

RESPONSE:

Thank you, we have edited the manuscript accordingly.

Comment:

Figure 1 is confusing. It seems that all three steps are just half of the way of a SLR as the graphic stops with half a circle. Why? Just for graphical reason?

RESPONSE:

We have edited the Figure, please see the edited version of the manuscript.

Comment:

Regarding the search terms: why was “Islamic” and not “Islam*“ used - with this approach you would neglect studies that e.g. use the term “Islam“. Same goes for „sustainable development “why not “sustainab” to include „sustainability“? Please justify or adjust.

RESPONSE:

The search explored many key words including Islam, models, practices and SDGs but the outcome was very limited. After many trials and testing a final selection of these key words were concluded.

Comment:

Why was only WoS used as a database? Ok it is a very comprehensive one, but a good review should never just rely on one database. Include others at least for triangulation of results now.

RESPONSE:

Triangulation using other data based was used including Science Direct, Pro Quest Central, and Google Scholar to help in the discussion and analysis and future directions. See text in yellow to address this point.

Comment:

The protocol can be put into the appendix.

RESPONSE:

We moved it to Appendix 1. (please see attached).

Comment:

According to your sampling funnel, 31 studies could not be found or were inaccessible - this is ca.. 15% of the identified studies in the first step. Maybe that can be reduced through other databases as recommended before.

RESPONSE:

For the sampling funnel 8 studies could not be found or were inaccessible. This is lower than 15% of the identified studies in the first step.

These studies were categorized as either irrelevant or in different languages (other than English) or a local conference. However, other data bases were reviewed like Science Direct and EBSCO to ensure better coverage and triangulation.

Comment:

What about cross-referencing? At the moment there is no details on this and I fear that some relevant studies are left out due to the too narrow focus on one database only.

RESPONSE:

Your point is well taken and sound. Please note that other data bases were reviewed like Science Direct and EBSCO to ensure better coverage and triangulation. (see text in yellow).

Comment:

A major point is the delineation from existing literature reviews in the field (e.g. Islamic banking sustainability: A review of literature and directions for future research (2017) or Energy economics in Islamic countries: A bibliometric review (2021)). There should be a paragraph summarizing existing findings of those review studies and how they differ from your study. This would also underline and strengthen the need for the study which I recommended earlier.

RESPONSE:

Point well taken, please note that we reviewed these two papers and included in the references and were commented upon (see text in Yellow).

Comment:

Now there are way too many tables in the manuscript which disturbs the reading flow heavily. I would suggest to streamline and put many of them into an appendix or delete.

RESPONSE:

Point well taken, a Table for drivers was moved to Appendix B.

Reviewer 2 Report

Although tue paper could be an interesting work, I am greatly concerned about the number of coincidences with other texts, exposed after the review.

The policy and style rules of "Sustainability" are clear and it is important that this type of situation does not occur. 

Author Response

Dear Editors

We are deeply grateful to the reviewers for taking the time to review our manuscript. We sincerely appreciate all valuable comments and suggestions which helped us to improve the quality of the article. We have given below our responses and explanations point by point for all the comments and suggestions given by the respected reviewers.

We hope that our manuscript will be acceptable for publication in Sustainability.

Respectfully,

Evren Tok, PhD

Response to Reviewer II

Comment:

Although the paper could be an interesting work, I am greatly concerned about the number of coincidences with other texts, exposed after the review.

The policy and style rules of "Sustainability" are clear, and it is important that this type of situation does not occur. 

RESPONSE:

Dear respected reviewer, thank you for your comments. Please note that this paper was submitted TWICE, and this may be the reason for this co-incidence. In addition, we considered the policy, guidelines, and rules of the Journal while editing the manuscript. We take into account the specific requirements regarding to the front matter, literature review sections, and the back matter.

Reviewer 3 Report

This is an interesting paper on islamic view in sustainable development debates.

It contributes to a gap in knowledge.

I would suggest a better transition between section 1 and 2. How do you relate methods to the concepts of the introduction?

Correct the graph (see figure 1 IndentificationN) on page 5. 

You should end section 2 with final considerations and transition to the descriptive analysis in the 3rd section.

A better discussion of the various tables is expected. 

In table 10 and following I would expect a reference to the sustainability models introduced on page 16.

Author Response

Dear Editors

We are deeply grateful to the reviewers for taking the time to review our manuscript. We sincerely appreciate all valuable comments and suggestions which helped us to improve the quality of the article. We have given below our responses and explanations point by point for all the comments and suggestions given by the respected reviewers.

We hope that our manuscript will be acceptable for publication in Sustainability.

Respectfully,

Evren Tok, PhD

Response to Reviewer III

Comment:

I would suggest a better transition between section 1 and 2. How do you relate methods to the concepts of the introduction?

RESPONSE:

Dear respected reviewer, we have edited the manuscript accordingly.

Comment:

Correct the graph (see figure 1 Identification) on page 5. 

RESPONSE:

Thank you, we have edited the manuscript accordingly.

Comment:

You should end section 2 with final considerations and transition to the descriptive analysis in the 3rd section.

RESPONSE:

Dear respected reviewer, thank you for the comments, we have edited the manuscript accordingly.

Round 2

Reviewer 1 Report

Dear authors,

thanks for working and on my comments and adjusting the manuscript accordingly. Great job!

Best,
Reviewer

Reviewer 2 Report

No comments